# Unravelling the neural signatures of dream recall in EEG: a deep learning approach

## Abstract

Dreams and our ability to recall them are among the most puzzling questions in sleep research. Specifically, putative differences in brain network dynamics between individuals with high versus low dream recall rates, are still poorly understood. In this study, we addressed this question as a classification problem where we applied deep convolutional networks (CNN) to sleep EEG recordings to predict whether subjects belonged to the high or low dream recall group (HDR and LDR resp.). Our model achieves significant accuracy levels across all the sleep stages, thereby indicating subtle signatures of dream recall in the sleep microstructure. We also visualized the feature space to inspect the subject-specificity of the learned features, thus ensuring that the network captured population level differences. Beyond being the first study to apply deep learning to sleep EEG in order to classify HDR and LDR, guided backpropagation allowed us to visualize the most discriminant features in each sleep stage. The significance of these findings and future directions are discussed.

## 1 Introduction

Dreams are arguably one of the most intriguing phenomena in cognitive science. Previous studies have explored the differences in brain EEG patterns between HDR and LDR using different approaches such as sleep spindles [1] and Event-Related Potentials [2] to identify the neural bases of dreaming. Although these findings have advanced the field in understanding the neural bases of dreaming, neural correlates of dream recall are still not completely understood.

In recent years, deep learning (DL) has proven to be one of the most successful techniques for classification problems, with specific applications in computer vision, natural language understanding as well as EEG signal decoding [3]. DL methods allow the identification of optimal discriminative patterns in a given minimally pre-processed dataset, thus reducing the reliance on *a priori* selection of features. However, the interpretability of such deep models has been a major roadblock in the context of neuroimaging applications.

In this work, we used a convolutional neural network (CNN) for classification of sleep EEG recordings into two groups (HDR vs LDR). Subsequently, we explored various techniques to visualize the features learned by the network.

## 2 Materials and Methods

### 2.1 Data Collection

The sleep study consisted of 36 participants (18 male, mean age $23 \pm 3$ yrs). Brief interview was conducted to determine which dream recall group the subject belonged. Those who reported more

that three dream recalls per week were classified as HDR and those who reported less than twice a month were categorized as LDR. Participants were made to sleep in an acoustically dampened and electrically shielded room.

Twenty-one AG/AgCl scalp electrodes were manually positioned according to the extended International 10-20 System (Fz,Cz,Pz,FP1,F3,FC1,C3,T3,CP1,P3,M1,O1). The electrophysiological data (EEG, EOG, and EMG) were continuously recorded via a BrainAmp system (Brain Products GmbH, Germany) with an amplification gain of 12,500, a high-pass filter of 0.1 Hz and a sampling rate of 1000 Hz with an anti-aliasing low-pass filter.

## 2.2 Sleep stage scoring and Data Preprocessing

Sleep stages were scored visually by two sleep experts using Rechtschaffen and Kales guidelines. The data was then split into 5s segments, each labeled as one of the 5 different vigilance state (wake, rapid eyes movements sleep – REM sleep, sleep stage 1 – S1, sleep stage 2 – S2 and slow wave sleep – SWS).

For our analysis, the data was downsampled to 200Hz. It was also passed through a low pass filter to exclude frequencies greater than 40Hz. Each data point was considered as an array of $19(eeg channels) \times 1000(timepoints)$ [5 second segment with 200Hz]. The number of data points available for each sleep segment is mentioned in Table 1.

Table 1: Available Data

| Sleep stage | Data points |
| --- | --- |
| S2 | 61860 |
| SWS | 52548 |
| REM | 35322 |
| S1 | 15120 |
| AWA | 14568 |

## 2.3 Model Architecture and training

We developed a CNN architecture for classifying each data point into one of the two dream recall groups. The model architecture is shown in Figure 1. The network contains two parts – a feature extractor and a classifier.

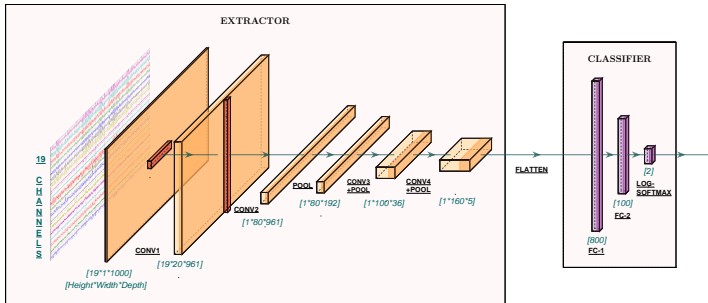

Figure 1: Model architecture

The preprocessed data (Section 2.2) is passed through the extractor (architecture described in 2). The first convolutional layer has a filter size of $1 \times 40$, which enables the network to capture temporal patterns of dream recall in each electrode. The second convolutional layer has a filter size of $19 \times 1$, which enables the network to capture spatial patterns across the recording electrodes. The output of the extractor is a 1-D feature vector of size 800, which is subsequently input to the classifier module (Figure 1). The classifier output represents log probabilities of the data sample to belong to the HDR or LDR classes.

The model was implemented using PyTorch 1.1 in Python. The negative log likelihood cost function was used to train the network and was then backpropogated through the classifier and extractor model. Adam optimizer was used for training and the learning rate was decreased using Step Scheduler. Early stopping was employed to prevent overfitting and the best model was stored for validation. The model hyperparameters used for training were Batch size: 100, Epochs: 150, Learning rate: 0.00005, Rate decay step size: 25 and Rate decay gamma: 0.7.

Table 2: Extractor network architecture

| Layer | Type | Filter Size | Output Size |
|---|---|---|---|
| 0 | Input | - | [1,19,1000] |
| 1 | Conv+BatchNorm | [1,40] | [20,19,961] |
| 2 | Conv+BatchNorm+Relu | [19,1] | [80,1,961] |
| 3 | MaxPool | [1,5] | [80,1,192] |
| 4 | Conv+BatchNorm+Relu | [1,5] | [100,1,184] |
| 5 | MaxPool+Dropout | [1,5] | [100,1,36] |
| 6 | Conv+BatchNorm+Relu | [1,10] | [160,1,27] |
| 7 | MaxPool | [1,5] | [160,1,5] |
| 8 | Flatten | - | [800] |

## 3   Results and Discussion

### 3.1   Decoding Accuracy

To test the model's cross subject accuracy, we used a leave 2-subjects-out strategy, wherein the model was trained on 34 subjects and tested on 2 held-out subjects (1 each from both dream recall groups). 18-Fold cross validation was done, and in each case 2 different subjects were a part of the testing set.

Figure 2 summarizes the decoding accuracy across the 5 sleep stages. We see that our deep model significantly outperforms the logistic regression model on all the sleep stages except S1 (might be attributed to the fact that less data was available for this stage). The model achieves decoding accuracy of around **70 to 75%** on the other 4 sleep stages.

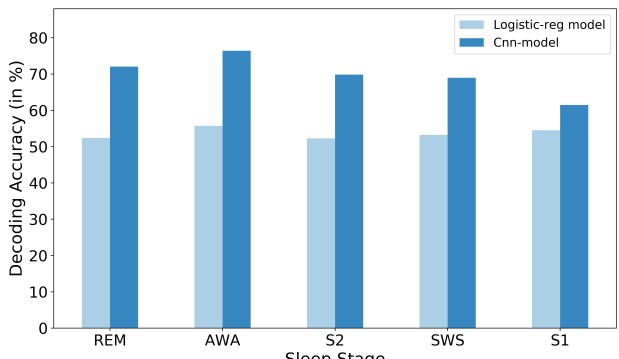

Figure 2: 18-fold cross validation accuracy compared across five sleep stages

### 3.2   tSNE Visualization to check subject overfitting

To assess the subject specificity of the features extracted by the network, we visualized a low-dimensional representation of the feature space using t-Distributed Stochastic Neighbor Embedding (t-SNE) [4]. To this end, we generated the tSNE plots for the output of the extractor part of the network, as shown in Fig. 3.

The left image corresponds to the feature space learned by our trained model. There were no visible clusters specific to a subject (each subject being represented by a color). We compared this plot to the feature space learned by a network trained to identify subjects from the EEG recordings (the right image). The formation of subject-specific clusters corresponds to the extracted features containing subject-specific information. This analysis confirmed that our proposed network did not learn features based on subject-specific information. Therefore, the learned feature space corresponds to population level differences between the two groups.

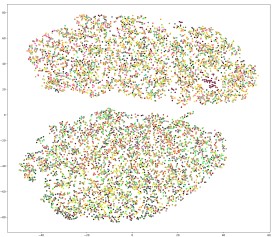 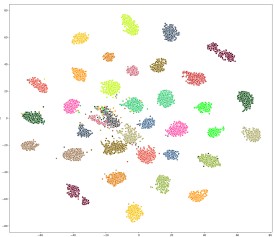

(a) Feature space learned by proposed CNN model

(b) Feature space learned by a model trained to identify subjects

Figure 3: tSNE embeddings calculated using outputs of feature extractor. Colours correspond to the subject label of that datapoint.

### 3.3 Electrode analysis using Guided Backpropogation

We used Guided backpropagation (GB) [5] technique to visualize the topological distribution of discriminative neural signatures of dream recall. Positive saliency map obtained using GB was used to infer the electrode importance. It was found that the electrode importance was similar across different validation sets for each sleep stage. The results obtained for various sleep stages are plotted in Figure 4.

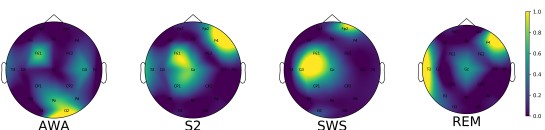

Figure 4: Electrode importance for dream recall classification as obtained from Guided-backpropagation. High intensity values correspond to higher importance given to that electrode.

## 4 Conclusions

This study illustrates how deep learning can be used to data-mine the neural activities of HDR and LDR. Specifically, we trained a deep CNN to classify between LDR and HDR and achieved significant decoding accuracies. Furthermore, we used tSNE to check for subject overfitting and GB to identify the brain regions carrying group-specific differences, thus illustrating the use of visualisation tools for deep models trained on neural data. Future work will involve explorations of other dimensions, including the frequency components of the data that contribute most to the classification.

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
