# OpenReview forum: "Unravelling the neural signatures of dream recall in EEG: a deep learning approach"
_NeurIPS.cc/2019/Workshop/Neuro_AI — Submitted to Real Neurons & Hidden Units @ NeurIPS 2019_

### Official Review · AnonReviewer3 · 2019-09-20
**CNN models provide a reasonably good classification of high vs low dream recall, but the insight on the neural mechanisms is minimal**

**Clarity:** 4

**Comment:**

You could look at features which are more directly related to brain activity and less influenced by mixing and volume conduction such as the shape of the waveforms, and the presence of bursts.

See also

Wong, W., Noreika, V., Móró, L., Revonsuo, A., Windt, J., Valli, K., & Tsuchiya, N. (2019). The Dream Catcher experiment: Blinded analyses disconfirm markers of dreaming consciousness in EEG spectral power. doi:10.1101/643593

for a set of measures which could be used.

**Category:**

AI->Neuro

**Clarity Comment:**

The paper is well written.

**Evaluation:**

2: Poor

**Importance:**

2: Marginally important

**Importance Comment:**

The paper describes the attempt of classification of a very specific and elusive feature, namely dream recall.
While in principle the effort could be important, there is no evidence of additional insight on the neuro/cognitive feature under exam.

**Intersection:**

3: Medium

**Intersection Comment:**

AI techniques are used to solve a neuroscience problem, but there is not enough evidence that neuroscience is involved or benefitted in this case.

**Rigor Comment:**

Nothing is said on data processing concerning artifact removal. In this sense we cannot be sure that the features responsible for the separability are related to neural activity or other (movement, etc).
Aso it's not clear how and why the groups were divided in "high" versus "low" dream recall, and what this means.
Also scalp signals contain a mixture of activity coming from different brain regions, on top of external signals, physiological artifacts etc. In this sense it's misleading to talk about brain signals, and even more of brain networks, in this context.

**Technical Rigor:**

2: Marginally convincing

---

### Official Review · AnonReviewer2 · 2019-09-24
**CNN binary classifier outperforms linear classifier on dream recall but with little neuroscientific insight**

**Clarity:** 4

**Comment:**

It would be interesting to see which features of the EEG signal contribute to the classification.

**Category:**

AI->Neuro

**Clarity Comment:**

The paper is easy to understand.

**Evaluation:**

2: Poor

**Importance:**

2: Marginally important

**Importance Comment:**

I think this is potentially a first step in an interesting direction, but without the ability to interpret which features of the EEG signal are important for classification there is little insight to be gained.

**Intersection:**

2: Low

**Intersection Comment:**

Though a deep learning model is fit to EEG data it does not (at the moment) teach us anything more about dream recall than the linear classifier.

**Rigor Comment:**

The model and training pipeline are well-devised. I would, however, like to see error bars on Figure 2 to understand to what effect these differences are significant.

**Technical Rigor:**

3: Convincing

---

### Official Review · AnonReviewer1 · 2019-09-27
**A neural network approach for EEG-based dream classification**

**Clarity:** 4

**Comment:**

Overall   comments:  The   experiments   conducted   do   not   necessarily   provide   a   strong   argument   that supports the  authors' claims.

Detailed comments:
Fig 1 - The description of the model architecture is a little confusing. Sticking with the convention of (depth/number of filters, width, height) would’ve worked better (as done in table 2).There is also an error in the text in figure 1, where depth and width have been interchanged.While the sizes of the filters have been mentioned, the number of filters used per layer haven't, explicitly. Given the errors in the figure’s text, thoroughly understanding the architecture is a little problematic. Also, the first two convolutional layers are described while the rest aren't, and the authors immediately move onto the output of the feature extractor, which is a little disconcerting.

Line 82 - Unclear what the authors mean by feature space. Would be a good idea to mention that these are the outputs of the feature extractor in the text as well as the caption of Fig 3.

Lines 90-91 - While it might be true that the feature space as defined by the author does not correspond to subject specific features, it should be noted that  these features are then further  passed through several fully   connected   layers.   It   is  possible  that   some  subject-specific   over-fitting   might   occur   in  the   deeper layers of the network. It might've been more convincing if the features at the last hidden layer had been visualized to make this point.

Fig 3 - In the same vein as the comment made about lines 90-91, it would've been interesting to see the t-SNE clusters coloured by HDR and LDR. Doing so would've also provided the reader with information if the network was learning features that allowed to cluster and therefore classify between the two classes of interest.

Lines 94-95 and Fig 4 - Do these visualizations provide us with any information? Does it make sense for these electrodes to be important for these sleep states? Guided   backprop   helps   say   which   features   are   important   for   a   particular   sample.   The   authors   fail   to mention which subject these visualizations belong to. Are they taken from a subject that's in the HDR or LDR   group?   Or   are   these   averaged   across   all   samples?  Are   there   any   differences   in   the   electrode importances between the two classes? Averaging the electrode importances for the two classes separately and showing them both for all sleep states would've been a more informative figure with respect to dream recall classification.

**Category:**

AI->Neuro

**Clarity Comment:**

The paper was relatively clear.

**Evaluation:**

2: Poor

**Importance:**

3: Important

**Importance Comment:**

Classification of neural states is an important problem.

**Intersection:**

2: Low

**Intersection Comment:**

Application of NNs to EEG data but little links back to neuroscience.

**Rigor Comment:**

Experiments weren't entirely convincing.

**Technical Rigor:**

2: Marginally convincing

---

### Decision · Program_Chairs · 2019-10-01

**Decision:**

Reject

**Comment:**

Unfortunately, we had more submissions than we could accept and based on the review process, we have decided not to accept your submission.  Nevertheless, thank you for your submission and interest in our workshop.